# The Worldwide Prevalence of Internet Addiction among Medical Students: A Systematic Review and Meta-Analysis

**DOI:** 10.3390/ijerph21091146

**Published:** 2024-08-29

**Authors:** Zhandos Salpynov, Zhanar Kosherova, Antonio Sarría-Santamera, Yerbol Nurkatov, Arnur Gusmanov, Yuliya Semenova

**Affiliations:** 1Department of Surgery, School of Medicine, Nazarbayev University, 010000 Astana, Kazakhstan; zhandos.salpynov@nu.edu.kz (Z.S.); zbeysenova@nu.edu.kz (Z.K.); 2Department of Biomedical Sciences, School of Medicine, Nazarbayev University, 010000 Astana, Kazakhstan; antonio.sarria@nu.edu.kz (A.S.-S.); arnur.gusmanov@nu.edu.kz (A.G.); 3Department of Medicine, School of Medicine, Nazarbayev University, 010000 Astana, Kazakhstan; yerbol.nurkatov@nu.edu.kz

**Keywords:** internet addiction, compulsive internet use, behavioral addiction, medical students, problematic internet use

## Abstract

Background: The internet helps us obtain necessary information, facilitates social communication, and provides access to entertainment content. The internet can also lead to the behavioral addictive condition termed internet addiction (IA) if used excessively. As active internet users, medical students are susceptible to IA, which is known to lead to depression and improper medical care delivery, poor academic performance, worse sleep quality, and undesirable financial issues. This systematic review and meta-analysis aimed to assess medical students’ pooled IA prevalence. Methods: The analysis included thirteen cross-sectional studies involving 4787 medical students. Cumulative, subgroup, and meta-regression meta-analyses were applied, using the random-effects model and the restricted maximum likelihood method. Results: The cumulative meta-analysis revealed a rise in the proportion of IA from 0.08 to 0.29, with minor fluctuations between 2015 and 2022. The IA prevalence in lower-middle-income countries was approximately three times higher than in high-income ones. Age and gender were not associated with IA among medical students. Conclusions: The worldwide prevalence of IA was 0.29, with a 95% CI between 0.19 and 0.41. Considering negative IA implications for medical students’ well-being, policymakers and all stakeholders should pay special attention to addressing IA within the medical student community.

## 1. Introduction

Due to rapid technological progress in the last ten years, the internet has become crucial to many individuals’ daily routines since it helps us obtain necessary information, facilitates social communication, and provides access to entertainment content. Despite the benefits of the internet, it can lead to a behavioral addictive condition known as “internet addiction” (IA) if used excessively [1]. Although it is commonly believed that the term addiction is only relevant to substances such as alcohol or nicotine, behavioral scientists claim that any stimulating agent could potentially be addictive [2]. When such behaviors as computer gaming and internet browsing become more than just habits and become compulsive and obligatory, this change might signify the development of addiction; IA is often viewed as a disorder of concern since it is accompanied with neural and cognitive dysfunctions (i.e., memory impairments), similar to those in substance and behavioral addiction disorders [3]. However, the main distinction is that a person is dependent on the online behavior itself and the associated emotional experiences, rather than on chemical substances [2]. In addition, IA frequently co-occurs with mental disorders, including depression [3], potentially suggesting its interplay with mental health. 

In 1995, I. Goldberg, an American psychiatrist, introduced IA to define pathological and compulsive online activity [4]. The term is also known as “pathological Internet use” [5], “problematic Internet use”, and “compulsive Internet use” [6]. K. Young, an American psychologist, uses IA to refer to a deterioration in the ability to control Internet usage, characterized by a range of cognitive, behavioral, and physiological symptoms [7]. As of now, there is a lack of agreement regarding the terms and definitions for IA. This has led to the frequent interchangeability of the above-mentioned terms in the academic literature [8], but the term IA can be applied to encompass the collective phenomenon [9]. It is worth noting that IA is not officially listed in the Diagnostic and Statistical Manual of Mental Disorders, Fifth Edition (DSM-5) [10]. However, the DSM-5 recognizes Internet Gaming Disorder (IGD) as a condition, which can be defined as “*the persistent and recurrent use of the Internet to engage in games, often with other players*”. It has been reported that IGD may also be labelled as “Internet Use Disorder”, “Internet Addiction”, or “Gaming Addiction” [11]. Furthermore, the recent confirmation by the World Health Organization (WHO) of IGD in the International Classification of Diseases 11th Revision (ICD-11) represents a significant milestone in addressing IA [12].

Seven specific areas should be assessed to determine if a person is addicted to the internet: tolerance; using the internet for a more extended period than planned; dedicating most of one’s time to internet usage; choosing virtual interaction over social and leisure activities of the real world; continuous engagement with the internet even when facing career, study, financial, and family-related issues; unsuccessful endeavors to control time spent on the internet; and withdrawal. If an individual experiences three or more of these areas, one can be considered internet-addicted [7]. Additionally, various psychological tests are used for assessing IA [13].

Researchers have identified several factors, contributing to individuals’ vulnerability to IA: social and cultural determinants (i.e., widespread access to the internet), natural propensity (i.e., genetic background), individual predisposition to mental health conditions, and others [13]. In other words, IA is a complex phenomenon driven by various factors [14], emphasizing the need to consider these factors for effective IA prevention and management [15].

The development and maintenance of IA can be determined by two theoretical models: Brand’s model and the model developed by Dong and Potenza [16]. According to the first model, specific personality traits (i.e., low self-esteem), psychological symptoms (i.e., depressive symptoms), and social perceptions (i.e., loneliness) predispose people to IA. Dong and Potenza’s model assumes that IGD is related to one’s attitudes and cognitive processes. The second model highlights the role of motivation-seeking behavior, which might have a direct influence on the onset of IGD [16]. As proposed by these two models, predisposing factors by themselves are insufficient to explain IA; rather, there are their interactions with individuals’ responses to stimuli associated with IA. While these models are theoretically feasible, they need further empirical examination [16].

IA can be observed in individuals of all ages today [17]. However, as active internet users, medical students are more susceptible to IA [1]. The Internet is frequently used for obtaining medical information [18], participating in social networking and other virtual activities [19]. Put differently, the Internet is indispensable for the medical community. However, its excessive use can cause negative consequences. Among medical students, IA is known to lead to depression and improper medical care delivery [18], poor academic performance [20], worse sleep quality [21], and undesirable financial issues [22].

Among medical students, the prevalence of IA varies both within and between countries. In India, 6% [5] to 58.8% were internet-addicted [23]. Ali et al. (2017) reported that 47.7% of medical students in Egypt experienced IA, while Ibrahim et al. (2022) observed a lower IA prevalence rate of 9.3% [18,22]. Elsewhere, 28.2% of medical students were internet-addicted [24]. The use of various IA tests and disparities in population characteristics could potentially have caused the observed variability in the results of studies [25].

During a literature review, it was found that studies on IA among medical students have mostly been limited to specific countries and used cross-sectional study methodologies, so there is a need for a systematic review and meta-analysis to combine the results of previously published international studies. Although Zhang et al. (2018) previously conducted a meta-analysis on IA among medical students, we believe that a new one is needed since the previous meta-analysis covered 10 studies from 2011 to 2017 [1]. Our meta-analysis includes 13 studies carried out between 2013 and 2023, ensuring a more comprehensive and current overview of the existing literature. Moreover, the COVID-19 pandemic has contributed to increased levels of anxiety and depression worldwide in all populations, as well as a higher incidence of IA. This rise in IA may be attributed to the necessary conditions introduced by COVID-19 of staying home and the greater use of online technologies for remote communication, work, and educational purposes [26]. Put another way, there is a need to update information about IA in medical students in the post-COVID-19 period. This will improve the precision and accuracy of estimating IA prevalence and its epidemiology, helping to draw robust conclusions about its impact on medical student populations. For those purposes, this systematic review and meta-analysis aimed to assess medical students’ pooled IA prevalence and analyze it through cumulative, subgroup, and meta-regression meta-analyses.

## 2. Materials and Methods

### 2.1. Registration, Study Design, Search Strategy, and Eligibility Criteria

The study was registered on the International Registration of Systematic Reviews (PROSPERO) platform (number CRD42023481454). The researchers applied a systematic review and meta-analysis study design, performing a comprehensive literature search from 1 December 2023 to 31 December 2023. The search strategy included combining the following search terms using Boolean logic operators: “Internet addiction” OR “Pathological Internet use” OR “Problematic Internet use” OR “Internet dependence” AND “Medical students.” Two independent researchers, Z.S. and Z.K., searched the following scientific databases: PubMed, the Cochrane Library, Science Direct, and Scopus. A supplementary literature search on the Google Scholar search engine was also performed to identify studies beyond these databases’ coverage, followed by manual reference tracking. For the conduct of the meta-analysis, we used the PRISMA 2020 checklist [27] (Appendix A).

The inclusion criteria were as follows: (i) only cross-sectional studies; (ii) articles written in English; (iii) only articles published between 2013 and 2023; (iv) articles with full-text availability; (v) medical students as study participants; (vi) studies used a validated questionnaire to measure the prevalence of IA; and (vii) sufficient data provided for pooled IA prevalence calculation. The exclusion criteria were (i) non-medical students as study participants, and (ii) studies not meeting the above-mentioned inclusion criteria.

### 2.2. Study Selection

At the initial stage of this study, all articles from various scientific databases were transferred to the Rayyan online platform. This online platform effectively screened abstracts and full-text studies, removing duplicate and irrelevant articles following inclusion and exclusion criteria [28]. Study titles and abstracts were screened at the next stage, and full-text articles were assessed. The study selection process is described in the Preferred Reporting Items for Systematic Reviews and Meta-Analysis flow diagram (PRISMA) depicted in Figure 1 [29].

### 2.3. Bias Risk Assessment

To appraise the methodological quality and reliability of the studies, the researchers simultaneously examined them using the Joanna Briggs Institute (JBI) critical appraisal checklist for prevalence studies [30]. The checklist includes nine questions that assess specific study domains for a potential risk of bias. JBI allows researchers’ responses to be categorized from 0 (no, unclear, and not applicable) to 1 (yes). A JBI score above 70% represents a low risk of bias. Scores of 50–69% reflect a moderate risk of bias, while less than 49% corresponds to a high risk of bias [31]. The systematic review and meta-analysis included only studies with moderate to low risk of bias. Z.S. and Z.K. disagreed regarding a study by Mohammadbeigi et al. (2016) [32]. Professor Y.S. decided that this study would be recognized as possessing a moderate risk of bias, so it was included in the analysis.

### 2.4. Data Extraction

Z.S. and Z.K. performed data extraction, with any discrepancies resolved by Prof. Y.S. From each article, data were extracted and meticulously inputted into a Microsoft Excel spreadsheet: (i) authors’ names and year of publication; (ii) country’s name and region; (iii) country’s income level; (iv) sample size; (v) study population characteristics (gender, age); (vi) IA data collection tool; and (vii) IA prevalence as an effect size.

### 2.5. Statistical Analyses

The inter-rater reliability was measured by calculating Cohen’s kappa statistic at the end of the study selection process. The recommended threshold values for Cohen’s kappa metric were as follows: less than 0.20, 0.21–0.40, 0.41–0.60, 0.61–0.80, and 0.81–1.00, reflecting slight, fair, moderate, substantial, and almost perfect or perfect agreement, respectively [33,34]. The calculation was performed using the free Cohen’s Kappa online calculator [34].

To justify the number of medical students included in this study, the Sample Size Calculator for Estimating a Single Proportion was used. The following input parameters were taken into account for sample size justification: level of confidence = 0.95; expected proportion = 0.3 (based on Zhang et al. (2018) [1]); and precision or margin of error = 0.05 [35].

The researchers used STATA software (Version 18) to perform statistical analyses grounded in the random effects (RE) model [36]. This model was applied to address statistical heterogeneity between studies [37]. The *I*^2^ statistic was employed to evaluate between-study heterogeneity. It offers a measure of the extent to which variability in results across studies can be attributed to real differences rather than random differences [38]. An *I*^2^ statistic of 25% shows a low heterogeneity, while a 50% score is attributable to moderate, and 75% refers to a high heterogeneity [1]. In addition, the *H*^2^ statistic was also employed to test the statistical homogeneity of the studies. If the *H*^2^ statistic equals 1, it indicates perfect homogeneity across the studies [39]. The restricted maximum likelihood (REML) method was used [40].

To estimate dispersion in effect size [41] and establish a plausible range for the effect size in a future study [40], we computed the 95% prediction interval (PI). The green whiskers extending from the overall diamond are used to represent the PI [40]. The effect size is presented as a logit-transformed proportion.

A cumulative meta-analysis (CM) was applied to accumulate evidence about IA prevalence. We started computing meta-analysis summaries for the first study, followed by the sequential inclusion of each subsequent study one at a time [39]. CM allows us to observe the dynamic shifts in IA prevalence as studies are gradually incorporated into the dataset [39]. Subgroup analysis was employed to explore how categorical variables might affect the prevalence of IA [1]. Meta-regression (MR) is a broader extension of subgroup analysis that examines the relationship between study effect size and covariates [40]. Constructing the forest plot allowed the research team to summarize the meta-analysis results.

Publication bias refers to the preference to publish studies with statistically significant results over those with non-significant ones [42]. To detect the bias, we constructed a funnel plot. A symmetrical funnel plot indicates no publication bias, while an asymmetrical one suggests its presence [40]. The number of missing studies that may exist because of publication bias was estimated in the nonparametric trim-and-fill analysis [40]. Additionally, Egger’s regression test was applied to quantify publication bias [43].

We performed the leave-one-out analysis (LOA) to examine each study’s influence on the overall effect size estimate [40]. The logit proportion was used in the LOA.

## 3. Results

### 3.1. Study Selection, Sample Size, and Risk of Bias Assessment

After identifying 713 studies, we removed 47 duplicate studies, resulting in 666 studies. Out of these 666 studies, 653 were excluded. Eight studies were excluded owing to a high risk of bias, while another eight had a low risk. Four of the remaining studies had a moderate risk (Appendix A). The interrater reliability was high (Cohen’s kappa statistic = 0.926), indicating almost perfect agreement in the meta-analysis’s judgments for study inclusion or exclusion.

According to the Sample Size Calculator for Estimating a Single Proportion, if 30% of the medical students suffer from IA, a sample size of 323 would be needed (Appendix A). Our analysis included 13 cross-sectional studies involving 4787 medical students. Thus, our sample size not only met but exceeded the necessary threshold.

### 3.2. Overview of Included Studies

Table 1 illustrates the characteristics of the included studies. This review examines studies from the following countries: India, Mexico, Iran, Egypt, Pakistan, Saudi Arabia, and Tanzania. Across the studies, the mean age was in the range of 19.9 and 23.8 years (SD = 1.35). The studies were classified according to their country of origin based on the World Health Organization’s six regions [44] and the income level classifications of countries by the World Bank [45].

### 3.3. Cumulative and Subgroup Meta-Analyses

As shown in Figure 2, the studies in the cumulative meta-analysis were sorted by their year of publication. The statistical metrics for Chauhan’s study represented the overall effect size since they incorporated data from all 13 studies. The analysis demonstrated that just over one-quarter of the sampled medical students had IA, with a 95% confidence interval (CI) of [0.19, 0.41]. The graph shows a rise in the prevalence of IA from 0.08 to 0.29 in the studies, with slight fluctuations observed over time.

Figure 3 illustrates the results of a subgroup meta-analysis examining the prevalence of IA among medical students, classified by data collection tool. The subgroup meta-analysis revealed that the majority of the studies (n = 12) applied Young’s internet addiction test (YIAT) to diagnose IA among medical students [51]. Conversely, the study conducted by Haroon et al. (2018) [46] uniquely employed an alternative IA diagnostic tool suggested by Tao et al. (2010) [52]. The number of studies in the group using Tao’s criteria was insufficient so it was impossible to calculate the heterogeneity statistics for that group. The overall proportion of IA medical students in studies using Young’s IA test was almost one-third (95% CI) [0.21, 0.44]. The higher *I*^2^ and *H*^2^ statistics in the subgroup meta-analysis indicated significant heterogeneity among the effect sizes between the two groups, supported by the statistically significant result of the test of group differences (*p* < 0.001). The 95% PI for the effect size in the Young IA test subgroup was in the range of [0.21, 0.44].

As shown in Figure 4, different proportions of medical students were affected by IA across different world regions. Southeast Asian medical students had a higher proportion of IA (about half) compared to American (less than one-tenth), Eastern Mediterranean (less than one-third), and African (approximately one-third) students. A test of group differences showed statistically significant differences in the magnitude of effect sizes among these studies (*p* < 0.001). The 95% PI for the effect size in the Eastern Mediterranean subgroup was in the range of [0.14, 0.40].

Figure 5 presents the outcomes of a subgroup meta-analysis examining how a country’s income level may influence IA among medical students. Eleven studies addressed the issue among medical students in lower-middle-income countries. The proportion of medical students with IA in these countries was approximately three times higher than in high-income countries (0.34 versus 0.12) and four times higher than in upper-middle-income countries (0.34 versus 0.08). The 95% PI for the effect size in the lower middle-income subgroup was in the range of [0.23, 0.47].

### 3.4. Meta-Regression Analysis, Leave-One-Out Analysis, and Publication Bias Assessment

This meta-regression analysis sought to examine the association between medical students’ IA prevalence and their mean age and gender. The meta-regression analysis showed that there was no association between IA and mean age (unadjusted β = −0.293, 95% CI = −0.797–0.211, z = −1.14, *p* = 0.25), and male gender (unadjusted β = −0.543, 95% CI = −4.940–3.853, z = −0.24, *p* = 0.809). To calculate the prevalence ratio of IA with a one-year difference, we further performed the exponentiation of unadjusted βs. The prevalence of individuals affected by IA displayed a negative correlation with age, wherein an increase in age by one year led to a decrease in the prevalence by 0.75. With a one percent difference in gender proportion among male medical students, the prevalence of IA was approximately 0.58 times higher compared to females.

In the leave-one-out analysis, all effect sizes were close to the overall effect-size red line, with their CIs crossing the line (Appendix A).

A funnel plot was used to assess publication bias by plotting logit-transformed proportions against their standard errors, with seven studies on the left and six on the right, as shown in Appendix A. The results of a nonparametric trim-and-fill analysis (NTFA) indicate that only one study was subjected to imputation (observed studies = 13 and imputed studies = 1; logit proportion for observed studies = −0.907, 95% CI = −1.461–−0.352; logit proportion for observed and imputed studies = −0.787, 95% CI = −1.355–−0.219). Egger’s regression test result could also be indicative of publication bias (β1= −12.31, SE of β1 = 3.24, z = −3.80, *p* < 0.001).

## 4. Discussion

As per the cumulative meta-analysis, the proportion of medical students addicted to the internet had minor variability over the years. Differences between the studies could have caused this variability or random sampling error [53]. As more studies are incorporated into a meta-analysis, the overall effect sizes become more consistent, decreasing the likelihood of a new study introducing a completely different effect size [40]. Accumulating additional data leads to increased precision and a reduction in the width of the confidence intervals.

It is of utmost importance to take a cautious approach when drawing definitive conclusions regarding the impact of data collection methods on IA prevalence among medical students in this subgroup meta-analysis. The analysis included twelve studies that used Young’s internet addiction test as a tool for data collection [51], and only one study was found to have used an alternative data collection tool, specifically the IA diagnostic criteria by Tao et al. (2010) [52]. It is recommended to consider multiple studies in a subgroup analysis to ensure generalizable findings. Relying on only one study may lead to concerns about the validity of the results and may not provide sufficient evidence to support claims or recommendations.

Differences have been observed in the proportion of IA among medical students in various regions across the world. Studies conducted in India (the Southeast Asian region) have shown a higher proportion of IA than in other geographic areas, with the pooled prevalence of IA of 51% and a 95% CI [41%, 60%]. This higher prevalence might be due to different reasons, one of which is the association of IA with a poor quality of life [1]. In India, half of the population has been reported to have a poor quality of life [54]. Therefore, it can be assumed that a poor quality of life might contribute to a higher IA prevalence of IA in medical students in India. Nonetheless, it is worth investigating other factors that might potentially serve as triggers for IA. Such factors might include an individual’s mental health status, but they are not limited to it. For example, the highest IA prevalence estimates were observed in Indian medical students with severe stress, anxiety, depression, and insomnia [55].

Additionally, it is crucial to adequately represent studies from African and American regions to gain a better understanding of IA in those areas. Currently, there is only one study from each, while eight originate from the Eastern Mediterranean and three from Southeast Asia. The overall prevalence of IA is estimated to be 34% with a 95% CI [23%, 47%] in lower-middle-income countries, compared to upper-middle-income (8%) and high-income (12%) countries. IA is a contemporary phenomenon, and low- and middle-income countries might face challenges regarding resources and policies required to address IA effectively [56].

There was insufficient evidence to establish a significant association between age and gender and a higher IA prevalence. The findings of the meta-analysis corroborate an earlier meta-analysis by Zhang et al. (2018) [1], which also determined that age and gender are not significant predictors for IA in medical students. Although young people and students are commonly regarded as the most vulnerable to IA since they are active internet users, according to Ioannidis et al. (2018), there is a lack of research on middle-aged and older age populations and their vulnerability to IA [57]. Meanwhile, Devine et al. (2022) reported that older ones were less likely to develop IA than younger adults, contributing to debates about the relationship between age and IA [58]. Similarly, the impact of gender on being predisposed to IA remains a controversial topic with conflicting results from different international cross-sectional studies. Some studies have found no discernible gender differences among medical students [19,59], while others have reported that male medical students were more vulnerable to IA than females [18,20]. We believe that the variation in results regarding gender and IA can be determined by people’s habits and cultural values, internet availability, and the policies of institutions [60].

We compared the pooled prevalence of IA among medical students with other population groups to determine how IA affects different populations. The current global prevalence of IA in medical students (29%) was found to be approximately twice as high as that of the general population (14.2%) [61]. This general population included representatives of children (<12 years old), adolescents (12–18 years old), and adult populations (≥18 years old). Compared to the general population, the higher prevalence of IA among medical students might be explained by the following reasons. Firstly, IA can be caused by medical students’ extensive use of the internet for research, academic assignments, and participation in virtual lectures [17]. Secondly, it is known that depression and anxiety are prevalent conditions in medical students, with global prevalence rates of 27.2% for depression and 47.7% for anxiety [62]. The risk of IA increases if individuals actively use the internet to cope with depression and anxiety [63,64]. There is also some evidence that those addicted to the internet had a 14-fold higher risk of depression and a 3.3-fold increased likelihood of anxiety compared to non-addicts [17], suggesting that IA and depression/anxiety are interrelated. Thirdly, IA prevalence rates might vary because of inconsistencies in measurement methods, different cut-off thresholds, and discrepancies in sampling methods [63].

However, when comparing the prevalence of IA among medical students with that in high school and non-medical university students aged 14.15 to 24.4 in Africa [65], we found that the prevalence of IA was approximately 1.17 times higher in high school and non-medical university students compared to medical ones (34% versus 29%). The higher IA prevalence in African young people can be attributed to such determinants as unemployment leading to more free time and thus greater opportunities for internet use. Additionally, it might be partially because of differences in Africans’ occupations and family structures [66].

LOA demonstrated that no study disproportionately affected the overall prevalence estimate, indicating the robustness of our meta-analysis results. 

Based on the nonparametric trim-and-fill analysis, we identified the presence of publication bias in the funnel plot. The observed asymmetry in the funnel plot may be attributable, in part, to heterogeneity between studies [67]. Regarding Egger’s test, there is some evidence of the presence of publication bias and small-study effects. Smaller studies might be more likely to have larger or variable effect sizes compared to larger ones [68].

### Limitations, Practical Implications, and Future Research Suggestions

There are some limitations in the systematic review and meta-analysis. First, the included studies exhibited significant heterogeneity, with only a minor proportion attributable to medical students’ age. However, the underlying factors that contribute to the remaining heterogeneity remain unknown and necessitate further investigation. Second, the current study did not include information on such factors as the reason for internet usage, the percentage of individuals with access to the internet in each country, and participants’ quality of life. These factors could potentially serve as contributing factors to the prevalence of IA [1]. Third, the analysis only encompassed studies published in English, potentially introducing a language bias [69]. Fourth, causality between IA and being a medical student cannot be established because of the cross-sectional nature of the included studies [1]. Fifth, we excluded studies with a high risk of bias from the beginning and did not include them in the analyses. By excluding the studies, we hoped to produce more reliable evidence. However, making a clear-cut distinction between high- and low-quality trials is challenging, and we can never be certain if a study is biased; we can only suppose that the study might be prone to bias [70]. Sixth, the current study did not include materials from the gray literature for the meta-analysis; however, evidence suggests that all sources of gray literature can help to define and contextualize research phenomena, as it sometimes contains relevant information that may not be fully covered in research papers [71]. Seventh, we could not calculate the 95% PI for each subgroup in the meta-analysis because PI can only be computed if there are at least three studies in a subgroup [40]. We only calculated the 95% PI for the Young IA Test, the Eastern Mediterranean, and the lower-middle-income subgroups, and the overall effect size.

Understanding the IA-related prevalence and its impact on medical students can aid in the development of support systems at medical universities, such as mental health and stress management services. We believe that providing access to mental health professionals specifically trained to address IA can help medical students manage IA and develop healthier coping strategies.

This study’s findings suggest that more research is needed to determine the extent of IA prevalence among medical students and its underlying causes. With this in mind, we suggest conducting additional subgroup meta-analyses encompassing a broader spectrum of studies from high-income countries and employing alternative IA data collection tools. In addition, future meta-analytic studies should examine how the type of internet activity, social and mental health support, the presence of mental health problems in medical students, and ethnicity interrelate with IA. By doing so, the comprehensiveness, validity, and applicability of IA findings for medical students can be improved.

## 5. Conclusions

In conclusion, it has been observed that there are differences in IA prevalence among medical students in various regions worldwide. The current meta-analysis reported the worldwide prevalence of IA at 29%, which was comparable with the pooled IA prevalence of 30.1% reported in a prior meta-analysis by Zhang et al. (2018) [1]. IA among medical students was much higher in the Southeast Asian region than in other world regions. Lower-middle-income countries had a higher IA prevalence than upper-middle-income and high-income countries. The current evidence is insufficient to establish the association between medical students’ age, gender, and a higher IA prevalence. Considering negative IA implications for medical students’ well-being, policymakers and all stakeholders should pay special attention to addressing IA within the medical student community.

## Figures and Tables

**Figure 1 ijerph-21-01146-f001:**
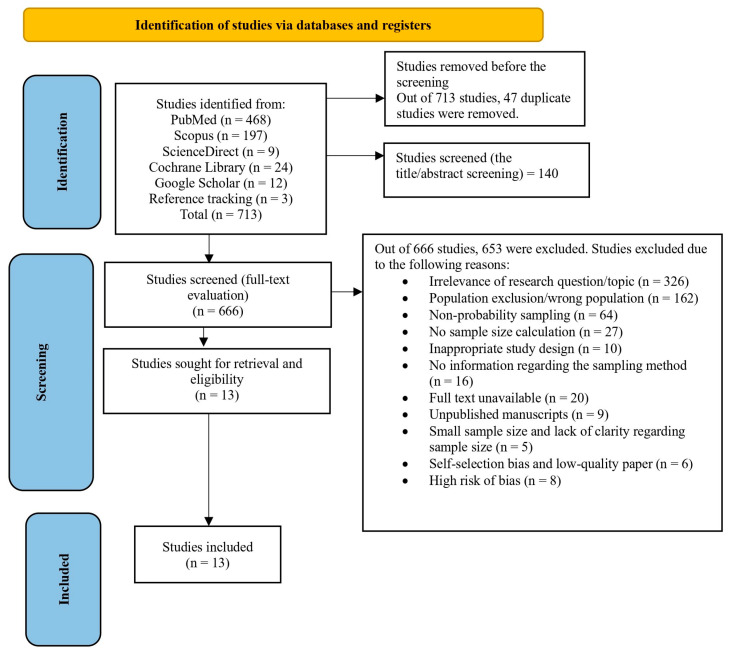
PRISMA 2020 flow diagram.

**Figure 2 ijerph-21-01146-f002:**
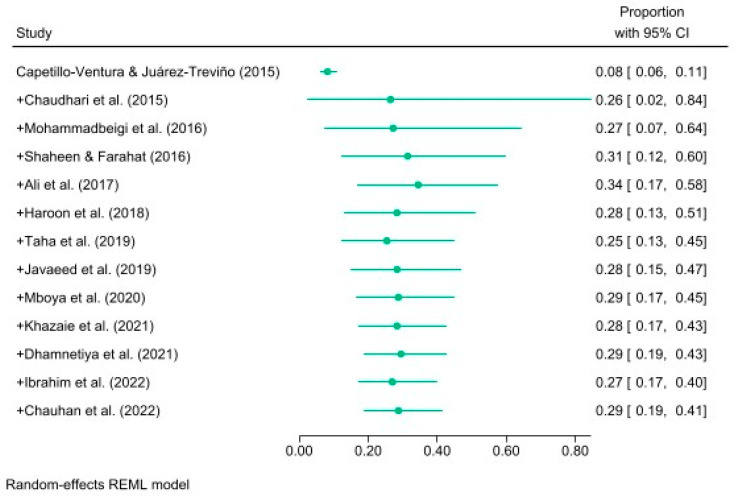
Forest plot of studies (n = 13) in a cumulative meta-analysis examining internet addiction prevalence by year of publication ([7,18,19,20,21,22,23,32,46,47,48,49,50]). Note. The “+” sign used in this analysis indicates the cumulative addition of each subsequent study to the previous ones.

**Figure 3 ijerph-21-01146-f003:**
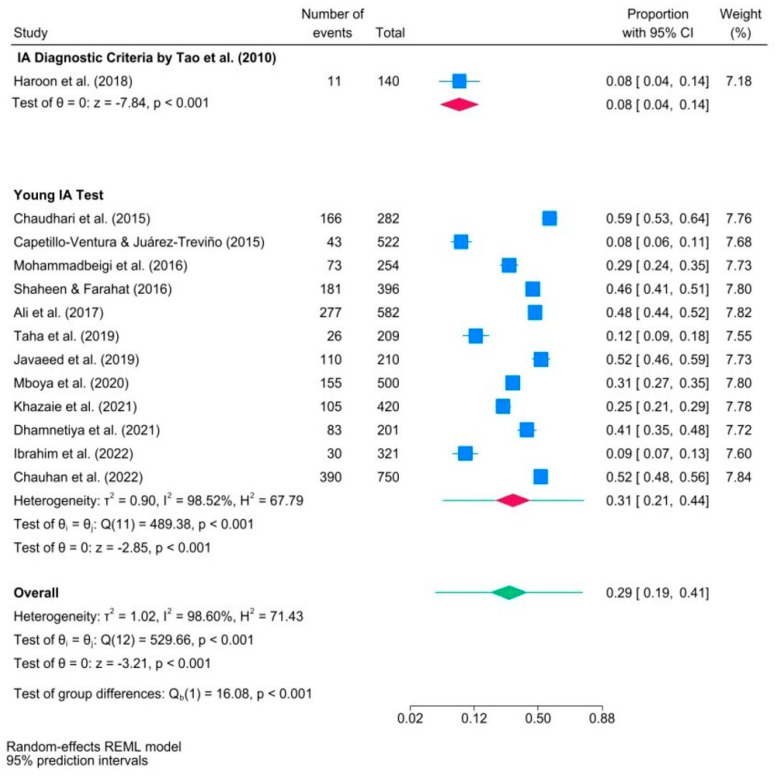
Forest plot of studies in subgroup meta−analysis examining internet addiction prevalence in medical students, categorized by data collection tools ([7,18,19,20,21,22,23,32,46,47,48,49,50]). Note. Blue squares indicate the prevalence estimates for each study. The red diamonds illustrate IA prevalence within subgroups, and the green diamond indicates the overall IA prevalence.

**Figure 4 ijerph-21-01146-f004:**
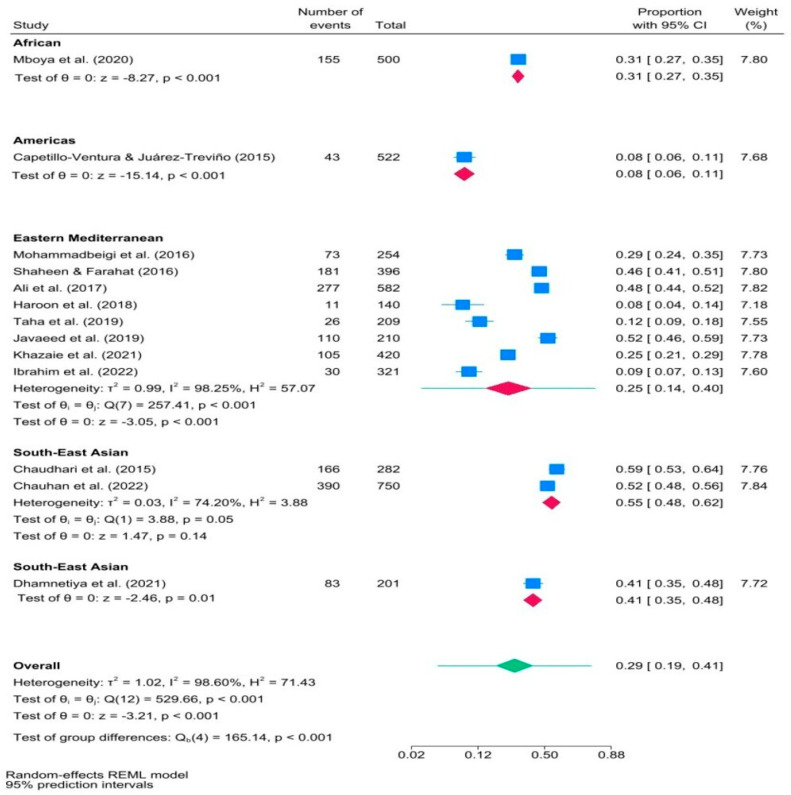
Forest plot of studies in subgroup meta−analysis examining internet addiction prevalence in medical students, categorized by world region ([7,18,19,20,21,22,23,32,46,47,48,49,50]). Note. Blue squares indicate the prevalence estimates for each study. The red diamonds illustrate IA prevalence within subgroups, and the green diamond indicates the overall IA prevalence.

**Figure 5 ijerph-21-01146-f005:**
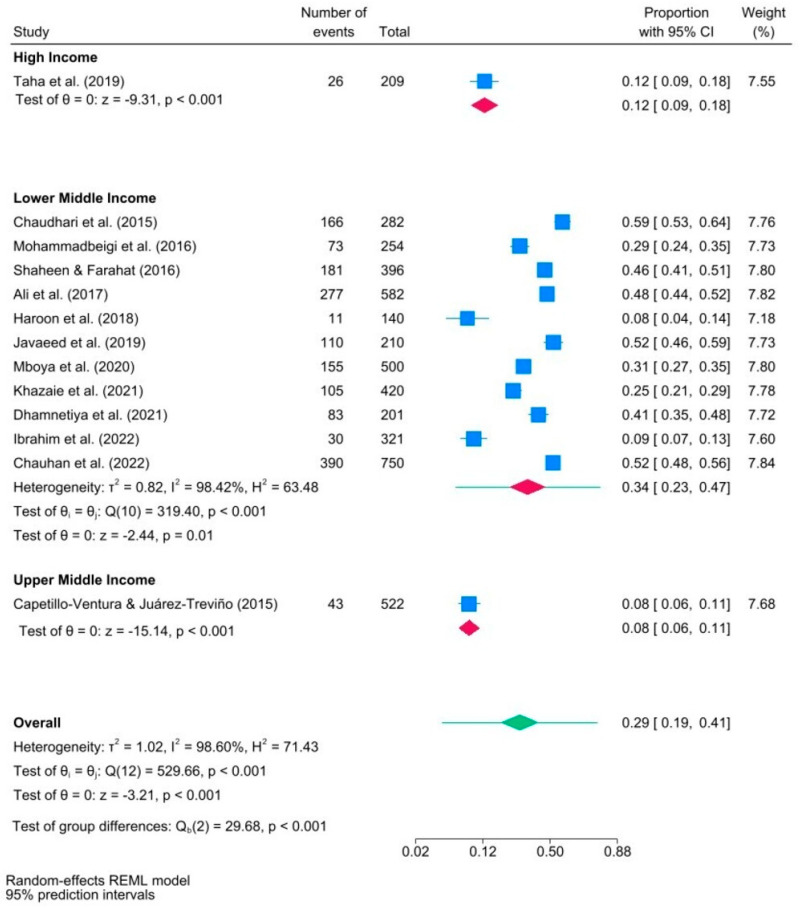
Forest plot of studies in subgroup meta−analysis examining internet addiction prevalence in medical students, categorized by country’s income level ([7,18,19,20,21,22,23,32,46,47,48,49,50]). Note. Blue squares indicate the prevalence estimates for each study. The red diamonds illustrate IA prevalence within subgroups, and the green diamond indicates the overall IA prevalence.

**Table 1 ijerph-21-01146-t001:** Characteristics of included studies.

Authors	Year of Publication	Country	Region	Country Classification by Income Level	Prevalence of IA	Sample Size	Mean Age	Data Collection Tool	Proportion of Males
Chaudhari et al. (2015) [23]	2015	India	Southeast Asian	Lower Middle Income	0.589	282	19.9	Young’s IA Test	0.432
Capetillo-Ventura and Juárez-Treviño (2015) [7]	2015	Mexico	Americas	Upper Middle Income	0.082	522	21.2	Young’s IA Test	0.538
Mohammadbeigi et al. (2016) [32]	2016	Iran	Eastern Mediterranean	Lower Middle Income	0.287	254	21.7	Young’s IA Test	0.160
Shaheen and Farahat (2016) [20]	2016	Egypt	Eastern Mediterranean	Lower Middle Income	0.457	396	19.7	Young’s IA Test	0.581
Ali et al. (2017) [22]	2017	Egypt	Eastern Mediterranean	Lower Middle Income	0.476	582	N/A	Young’s IA Test	0.349
Haroon et al. (2018) [46]	2018	Pakistan	Eastern Mediterranean	Lower Middle Income	0.079	140	N/A	Tao IA Diagnostic Criteria	0.485
Taha et al. (2019) [47]	2019	Saudi Arabia	Eastern Mediterranean	High Income	0.124	209	N/A	Young’s IA Test	0.578
Javaeed et al. (2019) [19]	2019	Pakistan	Eastern Mediterranean	Lower Middle Income	0.524	210	21.8	Young’s IA Test	0.357
Mboya et al. (2020) [48]	2020	Tanzania	African	Lower Middle Income	0.310	500	23.8	Young’s IA Test	0.584
Khazaie et al. (2021) [49]	2021	Iran	Eastern Mediterranean	Lower Middle Income	0.250	420	22.8	Young’s IA Test	0.466
Dhamnetiya et al. (2021) [50]	2021	India	Southeast Asian	Lower Middle Income	0.413	201	N/A	Young’s IA Test	0.656
Ibrahim et al. (2022) [18]	2022	Egypt	Eastern Mediterranean	Lower Middle Income	0.093	321	21.9	Young’s IA Test	0.455
Chauhan et al. (2022) [21]	2022	India	Southeast Asian	Lower Middle Income	0.520	750	20.3	Young’s IA Test	0.624

Note. N/A, not available.

## Data Availability

All the data presented in this study are included in the article, and further inquiries can be directed to the corresponding author.

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
