# Peer review of "The Worldwide Prevalence of Internet Addiction among Medical Students: A Systematic Review and Meta-Analysis"

_ijerph, 2024, doi:10.3390/ijerph21091146_

Round 1

Reviewer 1 Report

Comments and Suggestions for Authors

The evaluated manuscript is a meta-analysis of the prevalence of Internet addiction in medical students. The topic is highly relevant, and the findings could represent a significant advancement in understanding the impact of Internet addiction on current society, especially among the young.

Although the introduction mentions some aspects of Internet addiction, it is written superficially. It would be advisable to add more details and delve into certain key points:

For example, although terms like 'problematic internet use' or 'compulsive internet use' are mentioned, the use of the term 'addiction' is not justified, considering that it is not recognized as a disorder in either the DSM-5 or the ICD-11. Therefore, it would be advisable to justify its choice.

Likewise, using the terminology of the manuscript, the authors seem to suggest that simple excessive use of the internet leads to or can lead to addiction. I believe it would be advisable to clarify this aspect, as several recent studies disagree that merely prolonged or excessive use is a sufficient condition for addiction. These studies highlight the importance of considering other factors, such as underlying psychological issues or the context of internet use, in determining whether it constitutes an addiction.

Related to the previous point, theoretical models that explain the development and maintenance of Internet addiction are also not discussed,

Although it is mentioned in the discussion, it should be noted in the introduction that a previous meta-analysis on the same topic already exists and justify the need for a new one.

In relation to the methodology used, I believe the following aspects should be taken into account:

It is not mentioned whether the PRISMA checklist or another guideline was followed for the conduct of the meta-analysis.

In the selection of studies, it is mentioned that a screening was performed based on titles and abstracts, followed by a full-text selection. However, the flow diagram does not distinguish between these two phases and only presents a single evaluation of the criteria.

Regarding the evaluation of the methodological quality of the studies, it would be useful to include a table with the scores for each item of all the studies (even as supplementary material). Additionally, it is necessary to justify why studies with a high risk of bias were excluded, as they could have been included in the analyses and a sensitivity analysis conducted by removing them afterward.

It should be indicated who performed the data extraction process and the procedure that was followed. The resolution process for any discrepancies in data extraction should be reported.

Given that the I² index does not adequately reflect the heterogeneity of effects, especially in meta-analyses of prevalence, it would be advisable, as pointed out by IntHout et al. (2016), Migliavaca et al. (2022), and Borenstein (2023), to use the prediction interval to evaluate the degree of dispersion of true effects.

IntHout, J., Ioannidis, J. P., Rovers, M. M., & Goeman, J. J. (2016). Plea for  routinely  presenting  prediction  intervals  in  meta-analysis.BMJ Open,6(7), e010247.

Borenstein, M. (2023). How to understand and report heterogeneity in a meta-analysis: The difference between I-squared and prediction intervals. Integrative Medicine Research, 101014.

Migliavaca, C. B., Stein, C., Colpani, V., Barker, T. H., Ziegelmann, P. K., Munn, Z., ... & Prevalence Estimates Reviews—Systematic Review Methodology Group (PERSyst). (2022). Meta‐analysis of prevalence: I 2 statistic and how to deal with heterogeneity. Research synthesis methods, 13(3), 363-367.

No sensitivity analysis (e.g., leave-one-out) is conducted, which should be performed to assess the robustness of the obtained results.

It is not indicated whether any transformation (e.g., Logit) was used for the analysis.

The information on the analyses conducted to examine publication bias should be included within the statistical analysis section, rather than in a separate subsection. Additionally, I believe it would be beneficial to use other techniques to assess this aspect, such as Egger's regression test.

Similar to the introduction, the discussion lacks depth

It would be interesting to compare the prevalence found among medical students with the global prevalence as well as with that of other university students. Additionally, it could be explored whether internet addiction affects this specific population differently or more significantly.

I believe the limitations section of the study should be further developed, including among others, the omission of gray literature.

No practical implications or future research directions arising from the findings are established.

Author Response

Please see the attachment (Response to Reviewer 1)

Reviewer 2 Report

Comments and Suggestions for Authors
  • Abstract should be more concise and clearly convey the main objectives, methods, results, and conclusions.
  • Introduction should include a section distinguishing internet addiction from other types of addiction.
  • Different psychological tests should be included in the methodology to measure internet addiction, such as the Internet Addiction Test (IAT), Compulsive Internet Use Scale (CIUS), etc.
  • Present G*Power analysis results to justify the number of volunteers included in the study.
  • Expand the discussion to include a comparison with existing literature, ideally presented in a table format.
  • Add a subsection on the study's limitations and suggestions for future research.
  • Critically evaluate the statistical methods used and consider alternative approaches for more robust insights.
Comments on the Quality of English Language

 Minor editing of English language required

Author Response

Please see the attachment (Response to Reviewer 2)

Round 2

Reviewer 2 Report

Comments and Suggestions for Authors

The article may be accepted in its current form.

Comments on the Quality of English Language

Minor editing of English language required.